# Cross-cultural adaptation and content validation of the Infant Feeding Intentions Scale for Thai pregnant women

**Ratchanok Phonyiam**[1,2]*, **Donruedee Kamkhoad**[1,2], **Aunchalee E. L. Palmquist**[3,4]

**1** School of Nursing, University of North Carolina at Chapel Hill, Chapel Hill, North Carolina, United States of America, **2** Ramathibodi School of Nursing, Faculty of Medicine Ramathibodi Hospital, Mahidol University, Ratchathewi, Bangkok, Thailand, **3** Department of Maternal & Child Health, Carolina Global Breastfeeding Institute, Gillings School of Global Public Health, University of North Carolina at Chapel Hill, Chapel Hill, North Carolina, United States of America, **4** Carolina Population Center, University of North Carolina at Chapel Hill, Chapel Hill, North Carolina, United States of America

* ratch@ad.unc.edu

**Data Availability Statement:** All relevant data are within the paper and Supporting information files.

**Funding:** RP received the Alpha Alpha Chapter of Sigma Theta Tau International Research Grant to

## Abstract

This study aimed to validate a translated and culturally adapted version of the Infant Feeding Intentions (IFI) Scale for use in Thailand. Prenatal breastfeeding intention is a strong indicator of breastfeeding initiation. The stronger the intention to breastfeed among pregnant women, the more likely breastfeeding will be initiated after childbirth and continue for an extended period. There are currently no IFI scales that have been validated for use in Thailand. The translation of the IFI scale from English to Thai was conducted through a six-stage approach that included initial translation, synthesis of translations, back-translation, expert committee review for content validity, reliability testing, and submission of the translated IFI to notify the scale developers. Both Item and Scale Content Validity Indices equaled 1, scored by five experts, who also validated the content for cross-cultural adaptation. The final Thai IFI (T-IFI) scale demonstrated high content validity. A total of 30 Thai pregnant women participated in the reliability testing. The Cronbach's alpha of the 5-item T-IFI scale was 0.857, which indicated satisfactory internal consistency. The T-IFI scale demonstrated high content validity and was culturally appropriate for use in a Thai-speaking population. It has potential to strengthen assessments of prenatal infant feeding intention among pregnant women in Thailand.

## *Introduction*

Breastfeeding is among the most effective ways to promote both maternal and child health across the lifespan [1]. For women, breastfeeding is associated with improved health outcomes across the life course, including a reduced risk of breast and ovarian cancers, type 2 diabetes mellitus, and cardiovascular disease [2, 3]. Breastfeeding helps ensure the survival of infants and young children [4, 5]. Human milk is a complete source of nutrition, immune protection, and other bioactive proteins that contribute to infants' growth, cognitive development [4], and

complete this work. The funders had no role in study design, data collection and analysis, decision to publish, or preparation of the manuscript.

**Competing interests:** The authors have declared that no competing interests exist.

resistance to many infectious disease pathogens [4]. Breastfeeding also reduces the risk of a child becoming obese later in life [6]. The World Health Organization (WHO) recommends exclusive breastfeeding until six months, with continued breastfeeding and appropriate complementary foods up to two years of age or older [7].

Despite the robust evidence on established breastfeeding benefits, most infants are not exclusively breastfed from birth to 6 months or breastfed until at least two years of age as recommended [8]. Globally, only 25% are exclusively breastfed during the first six months [8] which is lower than the global target set by the WHO and the United Nations Children's Fund [9] to achieve at least 70% of children exclusively breastfed during the first six months by 2030 [9].

In Thailand, breastfeeding rates are even lower than the global average, although several interventions to promote and support breastfeeding have been implemented [10]. The Multiple Indicator Cluster Survey 2015–2016 reported that just 23.1% of infants under six months of age are exclusively breastfed [11]. A recent gap analysis in Thailand found that low breastfeeding rate was resulted from ineffectiveness of intervention in hospitals, communities, and workplaces [10]. A comprehensive strategy for achieving the breastfeeding target has not been yet established [10]. To achieve optimal breastfeeding practices, key recommendations are to increase investment in breastfeeding through family, health services, and workplace programs [10].

Prenatal breastfeeding education is one intervention that may improve both breastfeeding intentions and breastfeeding outcomes [12]. Prenatal breastfeeding intention is a strong indicator of breastfeeding initiation and adopting recommended exclusive breastfeeding practices [13]. Often, breastfeeding plans are made during pregnancy [13]. Evidence from multiple ethnic groups, included Asian, has shown that the stronger the intention to breastfeed, the more likely breastfeeding will be initiated and continue for an extended period [14, 15]. This relationship can be explained by considering that women who intended to breastfeed had consulted more sources of information about nutrition than women who did not intend to breastfeed ($p < 0.05$) [16]. Understanding the prenatal intention to breastfeed may help guide the development and implementation of public health policy as well as evaluation of interventions aimed at increasing exclusive breastfeeding rates. As such, valid and reliable tools are required to assess breastfeeding intention during pregnancy.

The Infant Feeding Intentions (IFI) Scale is a valid, simple tool for assessing the strength of intentions to initiate and sustain exclusive breastfeeding up to six months of age [14, 15]. The IFI scale has been validated in multiple ethnic groups in the U.S., including non-Hispanic white, African-American, and Asian expectant primipara women in the U.S. It has also been validated among English-speaking Hispanic and Spanish-speaking Hispanic expectant primipara women in the U.S. [14]. Thus far, the IFI scale has been translated into other languages for use in countries outside of the U.S., such as the Arabic version in Lebanon [17] and the Portuguese version in Brazil [18]. The IFI can be useful in addressing low breastfeeding rates by helping providers to understand breastfeeding intention among Thai women. However, to our knowledge, no Thai translations of the IFI were found, despite its internationally proven relevance. Therefore, the purpose of this study was to translate and culturally adapt the Infant Feeding Intentions Scale for pregnant women in Thailand.

## Materials and methods

This methodological and cross-sectional study was divided into two phases: 1) translation and cross-cultural adaptation of the IFI scale for Thai pregnant women and 2) quantitative assessment of its psychometric properties.

Written authorization to translate, adapt, and validate the IFI scale was granted by the developer of the original tool. Ethical approval was obtained from the University of North Carolina at Chapel Hill in the U.S. (IRB 21–1477) and Mahidol University in Thailand (IRB 3428). Written informed consent was obtained from all participants. All participants voluntarily participated in the study and was assured of anonymity and confidentiality.

### Instrument translation

The 5-item Infant Feeding Intentions Scale (IFI; Cronbach's alpha = 0.90) was developed to measure maternal prenatal intention to exclusively breastfeed [15]. The IFI scale uses a 5-point Likert- scale, individually scored from 0 to 4. The total IFI score is calculated by averaging the score for the first two items and summing this average with the scores for items 3–5. The possible score ranges from 0 to 16, with 0 representing a very strong intention not to breastfeed and 16 representing a very strong intention to exclusively breastfeed from birth until six months of age [15].

The translation of the IFI scale followed the translation process six-stage approach developed by Beaton and colleagues in 2000 [19]. Flowchart of the translation process is provided in Fig 1 illustrating the overall translation, cross-cultural adaptation, and validation process.

*Stage 1: Initial translation.* *Translation of the original English version of the IFI into Thai was independently performed by two bilingual Thai-English translators whose first language is Thai. The first translator (RP) is a PhD candidate with eight years nursing experiences on women's health (provided T1). The second translator (DK) is a PhD candidate with eight years nursing experiences on children's health (provided T2).*

*Stage 2: Synthesis of translations.* *This stage included both translators and a third researcher (AP), a Thai-American medical anthropologist, whose first language is English and heritage language is Thai. AP reviewed the translations and served as a mediator in the discussions of translation and cultural adaptation. Our team synthesized both T1 and T2 versions and developed a comprehensive Thai IFI (T3) before back translation.*

*Stage 3: Back-translation.* *Two back-translated versions were conducted on T3. The first was translated back into English by an English native speaker who did not have medical background or expertise in breastfeeding to avoid bias (provided BT1). Another version was back-translated by using the automated Microsoft Office translation program (provided BT2). Our research team then identified any concepts or terms that were inconsistent across versions. When dissimilarities between translators occurred, the original English version was referred to during the consensus process. A revision was made in T3 if needed.*

*Stage 4: Expert committee review for content validity.* *One final Thai IFI, hereafter T-IFI, along with previous versions (T1, T2, BT1, and BT2) were then reviewed by an expert panel comprised of five health care professionals who are active in women's health and breastfeeding research in Thailand. A cover letter and the T-IFI were included with the content validity form explaining the clear and concise instructions on how to validate each item. The panel was asked to report and describe any issues for each item with a recommended resolution in a write-up report. Then, RP had an individual meeting with each expert to discuss the content validity and a plan for revision. The first translator (RP) recorded experts' comments and suggestions in a written document. Adjustments were made after all five experts completed their reviews and provided feedback.*

For content validation, we used the Content Validity Index for each item and the overall scale [20]. The aim of this process was to analyze the degree of the agreement regarding the clarity and representativeness of the IFI, with answers obtained by a Likert scale and a score ranging from one (very unclear and needs full revisions) to four (very clear and does not need

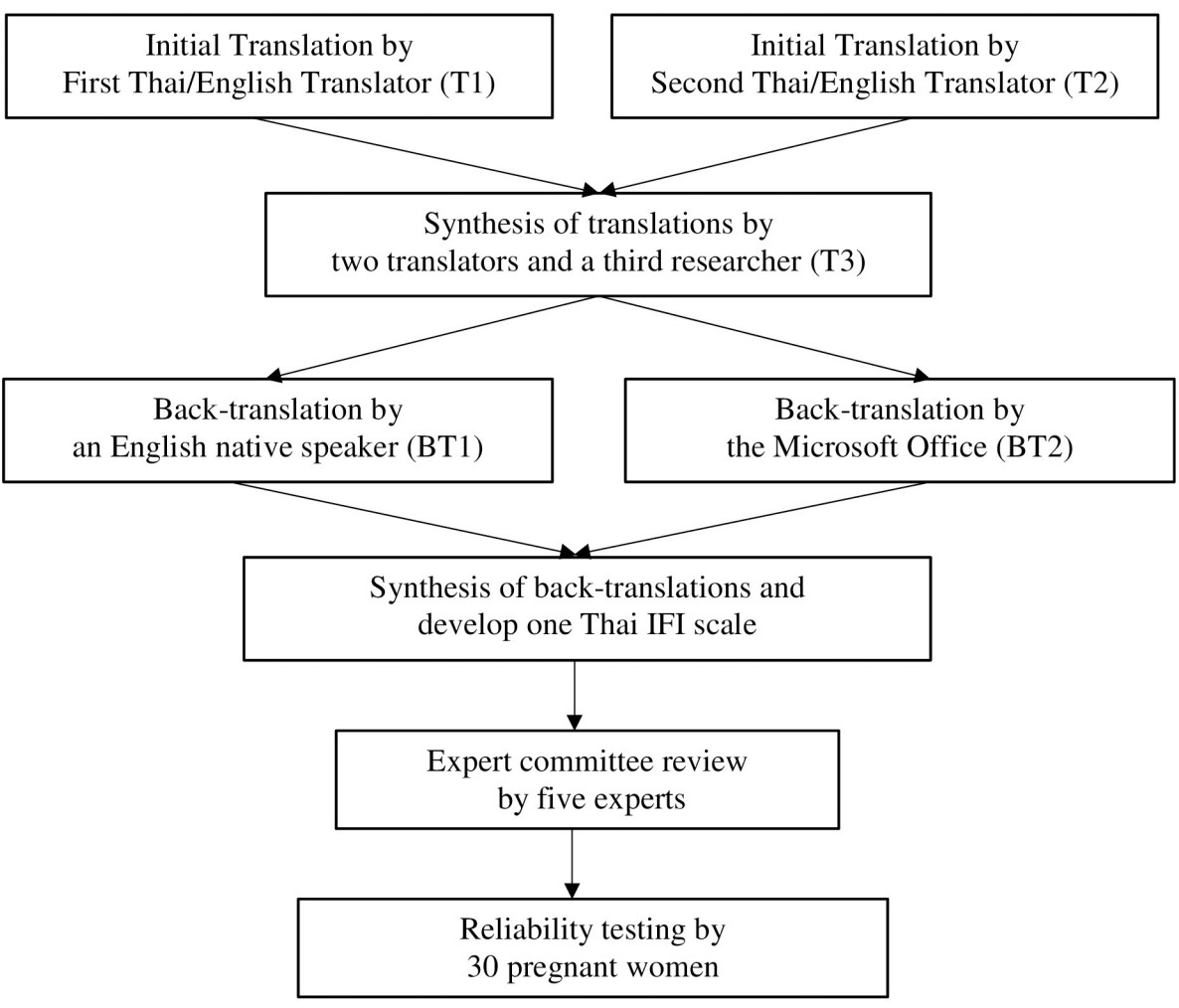

**Fig 1. Flowchart of the translation process.**

to be revised). Item Content validity Index (I-CVI) was calculated by the number of experts who rated this specific item as content valid (3 or 4 scores), divided by the total number of the experts. Scale Content Validity Index (S-CVI) was calculated by the number of items that were rated as content valid (3 or 4 scores) by the experts, divided by the total number of items [20].

Then, RP discussed the CVI score with each member of the expert review panel until consensus regarding the equivalence between the original and targeted versions to establish a pre-final version of the T-IFI. The expert panel assessed the pre-final version for clarity of the instructions, items' comprehensiveness, and cultural appropriateness. No further modifications were made. The T-IFI scale was then tested for its reliability.

**Stage 5: Reliability testing of the T-IFI scale.** *Following Beaton and colleagues (2000) guideline, a sample of 30 participants is considered sufficient for testing a scale [19]. Thai pregnant women eligible for inclusion in the study were 1) aged 20–44 years old, 2) without medical conditions (such as diabetes and hypertension), and 3) able to read, understand, and speak Thai.*

*Study participants.* Pregnant women were recruited from the antenatal care clinic of one hospital located in Bangkok, Thailand. Recruitments were achieved through face-to-face

outreach by a research assistant who worked as a registered nurse at the data collection site hospital. To recruit participants from various socioeconomic backgrounds, a research assistant randomly approached and handed out a flyer with the QR code. Pregnant women who were interested in our study used their smartphones to scan a QR code with a survey link. Participants then read information sheets and provided online written informed consent before participation.

From this survey link, pregnant women participated in an online survey that last approximately 15 minutes. They were asked to respond to 1) a demographic questionnaire that included questions about age, marital status, educational level, monthly household income, and employment status and 2) the 5-item T-IFI scale (see demographic questionnaire and T-IFI scale, S1 Text).

*Statistical analysis.* Data were entered and analyzed using IBM SPSS version 25.0 software [21]. Before we conducted the analysis, all data were screened for inconsistent or abnormal values. Continuous measures were assessed for normality and outliers. We calculated means, standard deviations, medians, and ranges for each continuous variable and tabulated frequencies and percentages for each categorical variable. SPSS software was used to analyze internal consistency reliability using Cronbach's alpha ($\alpha$) coefficient. A Cronbach's alpha value higher than 0.7 was acceptable [22].

**Stage 6: Submission of the T-IFI to notify the scale developers.** *We submitted the translation documents and the T-IFI scale to notify the scale developers.*

### Inclusivity in global research

Additional information regarding the ethical, cultural, and scientific considerations specific to inclusivity in global research is included in the (see, S2 Text).

## Results

The study's results provided preliminary support for the reliability and validity of the T-IFI scale which is culturally sensitive for Thai pregnant women.

### Cultural adaptation

There was no difficulty in translating the IFI scale from English into Thai. We found some discrepancies between the back-translated (BT1 and BT2) and the original scale.

First, the term "*baby*" used in the original version was translated into the same (*baby*) in BT1 but different in BT2 as the translation software used "*child*" instead. Our research team agreed that both *"baby"* and *"child"* were able to convey the same meaning. In this sense, *"baby"* is referred to *"เด็กทารก"* as a very young child while *"child"* is referred to *ªลูก"* or *ªเด็ก"* as a son or daughter of any age. Therefore, no changes were made because BT1 and BT2 confirmed that the Thai version was equivalent to the original version.

Second, the term *"any formula or other milk"* used in the original version was translated into *"formula or any other type of milk"* in BT1 and *"mixing or any other type of milk"* in BT2. We agreed that back-translated versions carry the same meaning as *"formula"* was referring to *"นมผสม"* in the Thai language. Therefore, no changes were made because BT1 and BT2 confirmed that the Thai version was equivalent to the original version.

### Content validity

The Content Validity Index [23] was scored by five experts. The CVIs for both each item and the entire scale were equal to 1 (see Table 1).

**Table 1. Content validity index by 5 experts.**

| Item | Expert 1 | Expert 2 | Expert 3 | Expert 4 | Expert 5 | I-CVI |
|---|---|---|---|---|---|---|
| 1 | 4 | 4 | 4 | 4 | 4 | 1 |
| 2 | 4 | 4 | 4 | 4 | 4 | 1 |
| 3 | 4 | 4 | 4 | 4 | 4 | 1 |
| 4 | 4 | 4 | 4 | 4 | 4 | 1 |
| 5 | 4 | 4 | 4 | 4 | 4 | 1 |
| S-CVI = 1 | | | | | | |

*Note*: I-CVI = Item Content validity Index was calculated by the number of experts who rated this specific item as content valid (3 or 4 scores), divided by the total numbers of the experts; S-CVI = Scale Content Validity Index was calculated by the number of items that were rated as content valid (3 or 4 scores) by the experts, divided by the total numbers of items.

After the expert panel review, our team agreed on adjusting one statement in items 3–5. In item 3, the English statement "*when my baby is one-month-old*" was changed in Thai to "*from birth to the first one month*" or "*ตั้งแต่แรกเกิดจนถึง 1 เดือนแรก.*" The same adjustment was applied to items 4 and 5, "*from birth to the first three months*" or "*ตั้งแต่แรกเดือจนถึง 3 เดือนแรก*" and "*from birth to the first six months*" or "*ตั้งแต่แรกเดือจนถึง 6 เดือนแรก,*" respectively. Our justification for the above adaptations was that rearranging items 3–5 helped carry more of a culturally meaningful statement about the duration judgment on breastfeeding.

Overall, the T-IFI scale demonstrated high content validity. The expert review committee validated the changes described above. Then, the revised T-IFI was then examined for reliability.

Before analyzing our qualitative data, we assessed for normality and outliers on continuous measures. The T-IFI scores had left-skewed distribution. Outliers were found from ones with higher income and those with lower T-IFI score.

**Table 2. Participant characteristics (n = 30).**

| Variable | % (n) | Mean ± SD | Range |
|---|---|---|---|
| **Age (years)** | | 32.23 ± 2.65 | 27–39 |
| **Marital Status** | | | |
| Single | 3.3% (1) | | |
| Marriage | 96.7% (29) | | |
| **Educational Level** | | | |
| Primary school | 3.3% (1) | | |
| Diploma | 6.7% (2) | | |
| Bachelor's degree | 66.7% (20) | | |
| Higher than bachelor's degree | 23.3% (7) | | |
| **Monthly Household Income (Baht)** | | 60,000 ± 64,490 | 0–300,000 |
| **Employment Status** | | | |
| Government/State enterprise | 50% (15) | | |
| Merchant/Personal business | 6.7% (2) | | |
| Private company | 23.3% (7) | | |
| Labor | 10% (3) | | |
| Freelance | 6.7% (2) | | |
| Others | 3.3% (1) | | |

*Note*: SD = Standard Deviation

### Participant characteristics

A total of 30 Thai pregnant women participated in a pilot assessment of the T-IFI (see, S3 Text). The demographic characteristics of the participants are presented in Table 2. Women were between the age of 27 and 39 years; the mean age was 32.23 years (SD = 2.65). Most women were married (96.7%). Two-thirds of women's educational level was the bachelor's degree (66.7%), followed by higher than bachelor's degree (23.3%). Monthly household income ranged from 0 to 300,000 Baht (SD = 64,490). Half of the women worked in government or state enterprises (50%), followed by in private companies (23.3%) and labor (10%).

### Internal consistency reliability

The Cronbach's alpha of the 5-item Thai IFI scale was 0.857 which indicated satisfactory internal consistency. Inter-total correlations ranged from .103 to .866. In case the item deleted, Cronbach's alpha ranged from .774 to .953, shown in Table 3.

## Discussion

This study aimed to translate, cross-culturally adapt, and validate the original Infant Feeding Intentions Scale [14, 15] from the English version into a Thai version, and then to test its psychometric properties. The study found that the T-IFI has appropriate internal and external content validity. This study suggested that this is an appropriate linguistic and cross-cultural adaptation of the T-IFI scale.

Following a standard translation approach [19], our translation and validation process showed the cross-cultural equivalence, since the IFI scale was translated by two bilingual translators and then synthesized by the third bilingual researcher who is knowledgeable in breastfeeding. Back-translation was conducted by one bilingual translator and another translation software. By using the software, we can assess the quality of wording that is generated during an automated literal translation without any background bias [24]. Our translation team compared and evaluated two back-translation sources to confirm accuracy. Throughout the translation process, our team was attentive to ensuring that the T-IFI so that it would be responsive to the ways that infant feeding is described in the Thai language. Throughout this cross-transcultural adaptation process, we found no difficulty in culturally adapting the IFI scale. In this regard, our study is consistent with other studies that have shown linguistic and cultural adaptability in multiple ethnic groups, including Asian populations in the U.S. [14].

The expert panel's review was used to refine the final translated T-IFI version. One adjustment we made was to modify the items' statements in terms of breastfeeding duration. The format of the T-IFI was the same as the original version, that was, 5 items with a 5-point Likert

**Table 3. Psychometric properties of the Thai Infant Feeding Intentions Scale (n = 30).**

| T-IFI | Mean (SD) | Corrected item-total correlation | Cronbach's α if item deleted |
|---|---|---|---|
| Item 1 | 3.43 (0.817) | .103 | .953 |
| Item 2 | 3.83 (0.747) | .799 | .800 |
| Item 3 | 3.67 (0.844) | .866 | .776 |
| Item 4 | 3.67 (0.844) | .866 | .776 |
| Item 5 | 3.47 (0.937) | .858 | .774 |
| T-IFI Cronbach's alpha = .857 | | | |

*Note*: T-IFI = Thai Infant Feeding Intentions; SD = Standard Deviation

scale. The results of content validity showed that the I-CVI and S-CVI have reached the recommendation standard [20]. Our T-IFI was pretested with a 30 sample of pregnant women in Thailand to help identify any difficulties with understanding or interpreting items. The findings suggested that the T-IFI had satisfactory reliability of Cronbach's alpha .857, indicating a homogenous concept on breastfeeding intention. Similar results have been reported in the original IFI scale tested with 64 Asian women [14], the Arabic version tested with 50 Lebanese women [17], and the Brazilian version tested with 31 Brazilian women [18]. Therefore, we were able to confirm that the T-IFI scale is not only suitable for use in the Thai cultural context but there is evidence for its validity and reliability.

In this study, the T-IFI total scores had left-skewed distribution which was similar to the original scale [14]. The T-IFI mean score was 14.43 out of 16, indicating pregnant women have a strong intention to breastfeed their children. Evidence has shown that Thai women who intend to breastfeed are more likely to seek various sources of prenatal recommendations [25] and cope well with unforeseen breastfeeding challenges [26]. Our T-IFI can be applied to screen Thai pregnant women about their intentions to breastfeed. In antenatal care services, health professionals may be able to tailor breastfeeding counseling and training sessions based on women's breastfeeding intention. Breastfeeding education can be delivered to facilitate women preparing to overcome their potential barriers in breastfeeding exclusively for the first 6 months of their baby's life.

Despite the satisfactory cross-cultural adaptation, our study has some limitations. First, our study had a small sample size of 30 pregnant women, who were mostly married and held at least Bachelor's degree education. Caution needs to be noted in the generalization of our findings to more demographically diverse Thai populations of pregnant women; therefore, further reliability and validity test studies in larger and more diverse population samples are recommended. Second, breastfeeding intentions were measured cross-sectionally during pregnancy only. Therefore, we were not able to assess whether the T-IFI predicted breastfeeding practices after childbirth. Further longitudinal research is needed to assess the ability of the T-IFI to predict postpartum intentions for early initiation and duration of exclusive breastfeeding among Thai women. Third, because this study was primarily conducted to confirm the reliability and validity of a new T-IFI scale, participants included both primigravida and multigravida women. We did not collect other attributes such as women's gravidity, parity, and previous miscarriages, which may have an impact on breastfeeding intention. We were also unable to completely exclude potential biases from unmeasured factors. Further studies to assess prenatal breastfeeding intentions using the T-IFI should collect more detailed information on previous pregnancy and infant feeding outcomes, which may help us better understand the mechanisms linking among multigravida women and further explore if/how primigravida and multigravida women breastfeed their babies differently according to their previous breastfeeding experiences. Last, our study was not able to conclude how pregnant women's demographics are associated with their intention to breastfeed the baby. Further research is needed to explore the associations between women's characteristics and their feeding intention during pregnancy.

## Conclusions

The translation and adaptation of the Thai Infant Feeding Intentions (T-IFI) Scale were successful. Satisfactory internal consistency reliability and content validity of the scale were demonstrated in this study. The need for further research is required to test T-IFI's construct validity across diverse Thai pregnant women. Consequentially, our T-IFI scale can be

applicable in clinical settings for Thai-speaking populations. It is as such recommended to assess a woman's infant feeding intention among Thai pregnant women.

## Supporting information

**S1 Text. Demographic questionnaire and T-IFI scale.**
(PDF)

**S2 Text. Checklist.**
(DOCX)

**S3 Text. Dataset.**
(XLSX)

## Acknowledgments

The authors would like to acknowledge Dr. Jiraporn Lininger, Dr. Jumpee Granger, Dr. Sangthong Terathongkum, Dr. Sopen Chunuan, and Dr. Anongnart Sirisabya for their expertise and for providing inside suggestions on content validity.

## Author Contributions

**Conceptualization:** Ratchanok Phonyiam.

**Formal analysis:** Ratchanok Phonyiam, Donruedee Kamkhoad.

**Funding acquisition:** Ratchanok Phonyiam.

**Methodology:** Ratchanok Phonyiam, Donruedee Kamkhoad, Aunchalee E. L. Palmquist.

**Resources:** Ratchanok Phonyiam.

**Supervision:** Aunchalee E. L. Palmquist.

**Writing – original draft:** Ratchanok Phonyiam, Donruedee Kamkhoad, Aunchalee E. L. Palmquist.

**Writing – review & editing:** Ratchanok Phonyiam, Donruedee Kamkhoad, Aunchalee E. L. Palmquist.

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
