## [Decision Letter · Decision Letter 0]

6 Aug 2022

PGPH-D-22-00762

Cross-Cultural Adaptation and Validation of the Infant Feeding Intentions Scale for Thai Pregnant Women

Dear Dr. Phonyiam,

Thank you for submitting your manuscript to PLOS Global Public Health. After careful consideration, we feel that it has merit but does not fully meet PLOS Global Public Health’s publication criteria as it currently stands. Therefore, we invite you to submit a revised version of the manuscript that addresses the points raised during the review process.

We look forward to receiving your revised manuscript.

Kind regards,

Julia Robinson

Executive Editor

Journal Requirements:

1. Please include a complete copy of PLOS’ questionnaire on inclusivity in global research in your revised manuscript. Our policy for research in this area aims to improve transparency in the reporting of research performed outside of researchers’ own country or community. The policy applies to researchers who have travelled to a different country to conduct research, research with Indigenous populations or their lands, and research on cultural artefacts. The questionnaire can also be requested at the journal’s discretion for any other submissions, even if these conditions are not met.  Please find more information on the policy and a link to download a blank copy of the questionnaire here: https://journals.plos.org/globalpublihealth/s/best-practices-in-research-reporting. Please upload a completed version of your questionnaire as Supporting Information when you resubmit your manuscript.

2. Please amend your detailed online Financial Disclosure statement. This is published with the article. It must therefore be completed in full sentences and contain the exact wording you wish to be published.

3. Please update your online Competing Interests statement. If you have no competing interests to declare, please state: “The authors have declared that no competing interests exist.”

4. In the online submission form, you indicated that “The datasets available from the corresponding author on reasonable request.”. All PLOS journals now require all data underlying the findings described in their manuscript to be freely available to other researchers, either 1. In a public repository, 2. Within the manuscript itself, or 3. Uploaded as supplementary information.

5. Please include a title page at the beginning of your manuscript file, that lists full author names and institute addresses. Do not upload as a separate file.  

6. Please provide separate figure files in .tif or .eps format and remove any figures embedded in your manuscript file. Please also ensure that all files are under our size limit of 10MB.

Additional Editor Comments (if provided):

Reviewers' comments:

Reviewer's Responses to Questions

**Comments to the Author**

1. Does this manuscript meet PLOS Global Public Health’s publication criteria? Is the manuscript technically sound, and do the data support the conclusions? The manuscript must describe methodologically and ethically rigorous research with conclusions that are appropriately drawn based on the data presented.

Reviewer #1: Yes

Reviewer #2: Yes

2. Has the statistical analysis been performed appropriately and rigorously?

Reviewer #1: Yes

Reviewer #2: Yes

3. Have the authors made all data underlying the findings in their manuscript fully available (please refer to the Data Availability Statement at the start of the manuscript PDF file)?

Reviewer #1: Yes

Reviewer #2: Yes

4. Is the manuscript presented in an intelligible fashion and written in standard English?

Reviewer #1: Yes

Reviewer #2: Yes

5. Review Comments to the Author

Reviewer #1: Comments:

Comment 1. The authors described in the title and objectives of the article that the "validation" of the instrument will be carried out, however the authors only perform the pre-test stage with 30 pregnant women, which characterizes only an initial stage of psychometric analysis or for some authors as the end of cultural adaptation. Therefore, I recommend that the authors change the title and objective of the work, indicating that the cultural adaptation of the instrument was carried out.

Comment 2. Regarding the calculation of the Content Validity Index, I recommend that the authors indicate the total reference value adopted for the index.

Comment 3. What is the reference adopted for the selection of 30 pregnant women in the pre-test stage? I recommend indicating the reference of Beaton et al., 2000, which indicates the inclusion of 30 subjects in this stage.

Comment 4. In the first paragraph of the results, the authors describe the sample number of the pre-test stage, however, to follow a logical sequence, I recommend that you start the Results topic with the description of the cultural adaptation stage.

Comment 5. At the conclusion of the study, the authors state "Satisfactory internal consistency reliability and content validity of the scale were demonstrated in this study. Consequently, our T-IFI scale can be applicable in research and clinical settings for Thai-speaking populations", however the scale has not yet been fully validated, so the authors should not make such a claim.

Reviewer #2: This manuscript describes a validation study conducted to validate a translated version of the Infant Feeding Intentions Scale adapted to the Thai context. The paper follows a rigorous methodology and is clear and concise in its presentation. Figure 1 is especially well done. The approach is an excellent template for other translations of the IFI for use in other contexts.

Limitations of the study are clearly articulated, which include limited demographic generalizability and lack of follow up data to enable validation of predictive value of Thai IFI. Thus, overall, this reviewer has only minor comments.

1. Table 1--consider a footnote to explain the calculation of I-CVI and S-CVI indexes. These are described in the text but is more informative when juxtaposed against the results summarized in Table 1.

2. In the results, consider first describing the participant characteristics and then presenting the ICR results.

3. The word 'Breastfeeding' with no qualifier (i.e., without 'exclusive' in front) is used throughout the manuscript when the original scale is designed to assess strength of EXCLUSIVE breastfeeding intention. Please clarify if the Thai adapted scale is designed to quantify any breastfeeding intention or exclusive breastfeeding intention? Was it an oversight or intentional to use 'breastfeeding' and not 'exclusive breastfeeding?'

4. What is the rationale for not including the translated scale as part of the manuscript?

6. PLOS authors have the option to publish the peer review history of their article (what does this mean?). If published, this will include your full peer review and any attached files.

**Do you want your identity to be public for this peer review?** For information about this choice, including consent withdrawal, please see our Privacy Policy.

Reviewer #1: **Yes: **Noélle O. Freitas

Reviewer #2: **Yes: **Laurie Ann Nommsen-Rivers

---

## [Decision Letter · Decision Letter 1]

2 Jan 2023

PGPH-D-22-00762R1

Cross-Cultural Adaptation and Content Validation of the Infant Feeding Intentions Scale for Thai Pregnant Women

Dear Phonyiam,

Thank you for submitting your manuscript to PLOS Global Public Health. After careful consideration, we feel that it has merit but does not fully meet PLOS Global Public Health’s publication criteria as it currently stands. Therefore, we invite you to submit a revised version of the manuscript that addresses the points raised during the review process.

EDITOR: Please insert comments here and delete this placeholder text when finished. Be sure to:

Thank you for submitting your manuscript to PGPH for publication. It has been reviewed by two independent reviewers. Although one recommended that the paper should be accepted, the other suggested minor revision. Their concern focuses on the sample size (I don't think this is an issue, as the study is not intended to be generalizable) and its calculation. I will suggest you adequately address their comments to avoid a delay in the publication of your manuscript. Also, read through the entire manuscript carefully to address any grammar, diction and structural issues. 

Thank you

Please ensure that your decision is justified on PLOS Global Public Health’s publication criteria and not, for example, on novelty or perceived impact.

Please submit your revised manuscript by 15/01/2023. If you need more time than this to complete your revisions, please reply to this message or contact the journal office at globalpubhealth@plos.org. Please include the following items when submitting your revised manuscript:

We look forward to receiving your revised manuscript.

Kind regards,

Dickson Abanimi Amugsi, PhD

Academic Editor

Journal Requirements:

Additional Editor Comments (if provided):

Reviewers' comments:

Reviewer's Responses to Questions

**Comments to the Author**

1. If the authors have adequately addressed your comments raised in a previous round of review and you feel that this manuscript is now acceptable for publication, you may indicate that here to bypass the “Comments to the Author” section, enter your conflict of interest statement in the “Confidential to Editor” section, and submit your "Accept" recommendation.

Reviewer #2: All comments have been addressed

Reviewer #3: (No Response)

2. Does this manuscript meet PLOS Global Public Health’s publication criteria? Is the manuscript technically sound, and do the data support the conclusions? The manuscript must describe methodologically and ethically rigorous research with conclusions that are appropriately drawn based on the data presented.

Reviewer #2: Yes

Reviewer #3: Partly

3. Has the statistical analysis been performed appropriately and rigorously?

Reviewer #2: Yes

Reviewer #3: No

4. Have the authors made all data underlying the findings in their manuscript fully available (please refer to the Data Availability Statement at the start of the manuscript PDF file)?

Reviewer #2: Yes

Reviewer #3: No

5. Is the manuscript presented in an intelligible fashion and written in standard English?

Reviewer #2: Yes

Reviewer #3: Yes

6. Review Comments to the Author

Reviewer #2: All reviewer concerns were addressed. I recommend acceptance of this manuscript.

Reviewer #3: The manuscript presents an important tool for testing the Infant Feeding Intentions for pregnant women towards breastfeeding for six months, which can be a useful tool for intention prediction.

In the materials and methods; the study is methodological and cross-sectional where the first one consist of translation

and cross-cultural adaptation of the IFI scale for Thai pregnant women and the later was a quantitative assessment of its psychometric properties.

The manuscript methodology is not well presented especially in the quantitative part.

1. How was the sample size calculated (i.e. 30 Women ?), what were the criteria used for the calculation?

2. How were these 30 participants found? which platform were used to select them? Does this account for any biasness in the results for a certain socioeconomic group?

3. The results part is also weak; there author indicated that they have used SPSS to test for normality and outliers; but in the results they have not presented this aspects.

4. Due to smaller sample size; the categories should be reduced and remove the ones with no observation (i.e. agriculture in the employment status.

5. What do the authors draw from the IFI and socio-demographic characteristics? Is there any story behind? Can the IFI be attributed by other factors among Thai Pregnant Women?

6. The results fails to justify the adaptation part of the IFI.

7. PLOS authors have the option to publish the peer review history of their article (what does this mean?). If published, this will include your full peer review and any attached files.

**Do you want your identity to be public for this peer review?** For information about this choice, including consent withdrawal, please see our Privacy Policy.

Reviewer #2: **Yes: **Laurie Nommsen-Rivers

Reviewer #3: No

---

## [Decision Letter · Decision Letter 2]

7 Feb 2023

PGPH-D-22-00762R2

Cross-Cultural Adaptation and Content Validation of the Infant Feeding Intentions Scale for Thai Pregnant Women

Dear Dr. Phonyiam,

Thank you for submitting your manuscript to PLOS Global Public Health. After careful consideration, we feel that it has merit but does not fully meet PLOS Global Public Health’s publication criteria as it currently stands. Therefore, we invite you to submit a revised version of the manuscript that addresses the points raised during the review process.

EDITOR: Please insert comments here and delete this placeholder text when finished. Be sure to:

Please ensure that your decision is justified on PLOS Global Public Health’s publication criteria and not, for example, on novelty or perceived impact.

We look forward to receiving your revised manuscript.

Kind regards,

Dickson Abanimi Amugsi, PhD

Academic Editor

Journal Requirements:

Additional Editor Comments (if provided):

Thank you for submitting your work to PGPH. Although one reviewer recommended that the manuscript be accepted for publication, another felt there are some minor issues that need to be addressed to make it publishable. I suggest you carefully addressed the reviewer's comments to enable me take a final decision on your manuscript. Also proofread it to address possible grammar/diction issues.

Reviewers' comments:

Reviewer's Responses to Questions

**Comments to the Author**

1. If the authors have adequately addressed your comments raised in a previous round of review and you feel that this manuscript is now acceptable for publication, you may indicate that here to bypass the “Comments to the Author” section, enter your conflict of interest statement in the “Confidential to Editor” section, and submit your "Accept" recommendation.

Reviewer #3: All comments have been addressed

Reviewer #4: All comments have been addressed

2. Does this manuscript meet PLOS Global Public Health’s publication criteria? Is the manuscript technically sound, and do the data support the conclusions? The manuscript must describe methodologically and ethically rigorous research with conclusions that are appropriately drawn based on the data presented.

Reviewer #3: Yes

Reviewer #4: Yes

3. Has the statistical analysis been performed appropriately and rigorously?

Reviewer #3: Yes

Reviewer #4: Yes

4. Have the authors made all data underlying the findings in their manuscript fully available (please refer to the Data Availability Statement at the start of the manuscript PDF file)?

Reviewer #3: Yes

Reviewer #4: Yes

5. Is the manuscript presented in an intelligible fashion and written in standard English?

Reviewer #3: Yes

Reviewer #4: Yes

6. Review Comments to the Author

Reviewer #3: All comments were addressed. On the final tone, please ensure proper grammar i.e. the use of "screwed", I think it meant "skewed".

Reviewer #4: PLOS Global Public Health Peer Review

Cross-Cultural Adaptation and Content Validation of the Infant Feeding Intentions Scale for Thai Pregnant Women

PGPH-D-22-00762R2

Thank you for the invitation to review this manuscript. This study is a Thai translation and cultural adaptation of the Infant Feeding Intentions Scale. Translating evidence-based questionnaires for LMIC and non-English speaking populations is a really important task. Given the need for quantifying, assessing, and addressing low breastfeeding rates in upper middle income settings like Thailand, this work has real utility and generalizability to other contexts. It would be great to test this in the future with a larger sample of Thai women. In general, this study is methodologically sound and should eventually be accepted. Below are suggestions for minor revisions.

This review is a second revision. To allow for line references for suggestions/comments/edits, this review refers to the track changes version of the document.

ABSTRACT

1. Line 72-74, “Content Validity Index was equal to 1” and “The expert review committee validated a cultural adaptation”: Could this be a bit more descriptive? “Both Item and Scale Content Validity Indices equaled 1, scored by 5 experts, who also validated the content for cross-cultural adaptation.”

2. Line 76, “culturally responsive” revise to “culturally appropriate” or “culturally adapted”?

INTRODUCTION

3. Line 100-101, “although several interventions…”: This paragraph does not explain the main factors why breastfeeding is low—what are the main factors? This may also give strength to the rationale/importance of this study—e.g., “the T-IFI is useful in addressing low breastfeeding rates by helping to understand breastfeeding intent”?

4. Line 106-108, “Prenatal breastfeeding education…”: perhaps this sentence is better to start the next paragraph.

5. Lines 111-113, “Evidence from the US…”: What does this mean for Asian populations? Thailand or southeast Asia?

6. Line 116 “hence further guides” revise to "may help guide"

MATERIALS AND METHODS

7. Lines 135-139, “Written authorization…”: Should this information be included at the end of the methods section? Should you include the official ethics IRB approval numbers?

8. Line 177 “PR”: RP or PR or is this an acronym for something else? If it is in reference to the corresponding author please make sure the correct initials in the rest of the document.

9. Line 181, revise “In the stages of content validation,” to “For content validation,….”

10. Line 196 “could be necessary” revise to “"a sample of 30 participants is considered sufficient for testing a scale". It seems that the authors have been asked this question about sample size from other reviewers. This answer is acceptable as the authors used a framework and mention in the Discussion that there is a need to test with a larger sample. The authors may also consider adding to the discussion how the sample size in this study compares to the original sample size in testing/validity of the IFI and how this might compare to other sample sizes for translations/cross-cultural adaptations of this IFI scale in other populations.

11. Line 207, “RA”: Some of these acronyms in the methods are a bit confusing, they mostly refer to the authors but now also to staff. In general, if I do not use a term that often, then I prefer to use the whole term instead of the acronym. I would consider this for "research assistant" in this manuscript.

12. Line 210, revise “handed in” to “handed out.”

13. Line 231, “Inclusivity in global research”: why is this mentioned here? Is this a requirement of the journal? If not, consider omitting.

DISCUSSION

14. Line 311, The study’s findings…”: Be careful of overstating the findings. Consider, "The study finds that the T-IFI has appropriate internal and external content validity. This study suggests that this is an appropriate linguistic and cross-cultural adaptation of the IFI survey."

15. Line 240, “more likely to access more sources” please revise.

16. Line 342, “In antenatal care services…”: This sentence is a little bit unclear. What is the utility of this scale in a clinical setting? To see who is interested in breastfeeding and tailoring education to them? To get pregnant women interested in breastfeeding? Can this scale aid in getting Thai women to breastfeed and helping Thai women overcome barriers to breastfeeding exclusively for the first 6 months of life?

17. Good job mentioning the need for caution and more reliability/validity testing in lines 347-350!

18. Line 353, “exclusive breastfeeding,” add “among Thai women.”

7. PLOS authors have the option to publish the peer review history of their article (what does this mean?). If published, this will include your full peer review and any attached files.

**Do you want your identity to be public for this peer review?** For information about this choice, including consent withdrawal, please see our Privacy Policy.

Reviewer #3: **Yes: **Adam Hancy

Reviewer #4: **Yes: **Ahmar Hashmi

---

## [Editor Report · Decision Letter 3]

23 Feb 2023

Cross-Cultural Adaptation and Content Validation of the Infant Feeding Intentions Scale for Thai Pregnant Women

PGPH-D-22-00762R3

Dear Phonyiam,

We are pleased to inform you that your manuscript 'Cross-Cultural Adaptation and Content Validation of the Infant Feeding Intentions Scale for Thai Pregnant Women' has been provisionally accepted for publication in PLOS Global Public Health.

Best regards,

Dickson Abanimi Amugsi, PhD

Academic Editor

Thank you for adequately addressing the reviewer's comments. Your manuscript is now suitable for publication in PGPH.